# Microfluidic Affinity Selection of B-Lineage Cells from Peripheral Blood for Minimal Residual Disease Monitoring in Pediatric B-Type Acute Lymphoblastic Leukemia Patients

**DOI:** 10.3390/ijms251910619

**Published:** 2024-10-02

**Authors:** Malgorzata A. Witek, Nicholas E. Larkey, Alena Bartakova, Mateusz L. Hupert, Shalee Mog, Jami K. Cronin, Judy Vun, Keith J. August, Steven A. Soper

**Affiliations:** 1Department of Chemistry, The University of Kansas, Lawrence, KS 66047, USA; mwitek@ku.edu; 2Center of BioModular Multiscale Systems for Precision Medicine, Lawrence, KS 66045, USA; hqs9cx@virginia.edu (N.E.L.); shaleemog@yahoo.com (S.M.); 3Department of Cancer Biology, The University of Kansas Medical Center, Kansas City, KS 66160, USA; 4Biofluidica Inc., San Diego, CA 92121, USA; alena.bartakova@gmail.com (A.B.); matt@biofluidica.com (M.L.H.); 5Division of Hematology/Oncology/Bone Marrow Transplant, Children’s Mercy Kansas City, Kansas City, MO 64108, USA; jkcronin@cmh.edu (J.K.C.); jvun@cmh.edu (J.V.); 6Bioengineering Program, The University of Kansas, Lawrence, KS 66045, USA; 7Department of Mechanical Engineering, The University of Kansas, Lawrence, KS 66045, USA

**Keywords:** B-type acute lymphoblastic leukemia, pediatric patients, minimal residual disease, affinity isolation, microfluidics

## Abstract

Assessment of minimal residual disease (MRD) is the most powerful predictor of outcome in B-type acute lymphoblastic leukemia (B-ALL). MRD, defined as the presence of leukemic cells in the blood or bone marrow, is used for the evaluation of therapy efficacy. We report on a microfluidic-based MRD (MF-MRD) assay that allows for frequent evaluation of blood for the presence of circulating leukemia cells (CLCs). The microfluidic chip affinity selects B-lineage cells, including CLCs using anti-CD19 antibodies poised on the wall of the microfluidic chip. Affinity-selected cells are released from the capture surface and can be subjected to immunophenotyping to enumerate the CLCs, perform fluorescence in situ hybridization (FISH), and/or molecular analysis of the CLCs’ mRNA/gDNA. During longitudinal testing of 20 patients throughout induction and consolidation therapy, the MF-MRD performed 116 tests, while only 41 were completed with multiparameter flow cytometry (MFC-MRD) using a bone marrow aspirate, as standard-of-care. Overall, 57% MF-MRD tests were MRD(+) as defined by CLC numbers exceeding a threshold of 5 × 10^−4^%, which was determined to be the limit of quantitation. Above a threshold of 0.01%, MFC-MRD was positive in 34% of patients. The MF offered the advantage of the opportunity for efficiently processing small volumes of blood (2 mL), which is important in the care of pediatric patients, especially infants. The minimally invasive means of blood collection are of high value when treating patients whose MRD is typically tested using an invasive bone marrow biopsy. MF-MRD detection can be useful for stratification of patients into risk groups and monitoring of patient well-being after completion of treatment for early recognition of potential impending disease recurrence.

## 1. Introduction

B-cell acute lymphoblastic leukemia (B-ALL) is an aggressive cancer characterized by the accumulation of immature white blood cells (i.e., blasts) in bone marrow and blood [1] and represents ~30% of cancers diagnosed in children [2]. The treatment of B-ALL pediatric patients consists of three stages: induction of remission, consolidation, and the maintenance of remission. The stages are designed to eliminate the leukemic cells and prevent disease recurrence. The kinetics of therapy efficacy (i.e., cancer cells response to treatment) is monitored via measurements of minimal residual disease (MRD), considered the most powerful predictor of outcome in acute leukemias, including B-ALL [3]. The limits of detection (LOD) of MRD assays determine the threshold level at which the patient’s disease is deemed MRD(−) or (+). Considering that the existing MRD detection assays (i.e., standard and next-generation MFC, PCR, NGS (next-generation sequencing), and microfluidic (MF)-MRD presented herein) offer different LODs, therefore, a better term describing MRD is “measurable residual disease” rather than “minimal residual disease” [4,5].

While vast numbers of B-ALL pediatric patients achieve complete remission following chemotherapy, unfortunately, ~10% experience a poor outcome due to disease relapse [6,7] owing to the presence of low levels of chemotherapy-resistant leukemic blasts [8] that are not detected by conventional MRD methods. Currently, MRD measurements are performed mainly on bone marrow aspirates (BMAs) via (i) microscopic evaluation of cells, (ii) MFC of cells, (iii) qPCR of fusion genes of nucleic acids, or (iv) NGS of gDNA [9,10,11,12,13,14]. Cytogenetic karyotyping (i.e., FISH) is used for the identification of leukemic cells with aberrant chromosomal characteristics but not for MRD assessment due to LOD [11].

As shown in Table 1, there are vast differences in analytical and clinical sensitivities among these MRD methods. Standard MFC is the primary method used for MRD testing [15] and can detect ~1 leukemia cell among 10,000 nucleated cells. MRD can be measured using PCR as well [16], providing higher sensitivity than MFC with the ability to detect 1 leukemia cell in 100,000 cells. However, PCR detects patient-specific targets, the identification of which are obtained based on DNA sequencing, which makes this test expensive and time consuming [11]. Additionally, for some patients, a unique PCR target cannot be identified or targets disappear during chemotherapy owing to the biological process [17], making this testing modality challenging for a wide population of patients. NGS shows the best analytical sensitivity [12,18,19]; it identifies more MRD(+) patients than MFC [19]. MRD in NGS is measured based on patient-specific blasts, probing VDJ, DJ IGH, and VJ TRG rearrangements. To identify leukemic cells’ specific sequence, a patients’ gDNA must be sequenced before treatment begins. The advantage of NGS is that it shows superior analytical sensitivity and low false-negative rates, provides benefits of identifying patients who are truly cured, those whose MRD is missed by MFC, and who should be potentially defined as high risk for relapse [19]. This is important, as one way to improve outcomes in B-ALL patients is to identify patients that have a high risk of impending acute relapse [20,21,22,23,24,25,26].

Considering all the MRD assays, we were motivated to develop one that will address the challenges of the existing methods and fulfill the following merits: (i) does not require BMA for MRD but can be performed from peripheral blood, (ii) does not need blood fractionation or pretreatment, (iii) can secure data frequently and provide results within hours, (iv) does not need diagnosis-specific information about the B-ALL phenotype or genotype to perform follow-up testing, and (v) can provide a better LOD and clinical sensitivities than standard MFC.

We report a microfluidic-based assay for monitoring of MRD from the blood of pediatric patients via affinity enrichment and immunophenotyping of circulating leukemia cells (CLCs). The cell selection microfluidic was used in our past work for the isolation of rare circulating tumor cells (CTCs) [27,28,29], plasma cell disorders (i.e., multiple myeloma, MM) [30], and acute myeloid leukemia (AML) [31], where we demonstrated the utility of the selection device for highly efficient cell isolation directly from peripheral blood, as determined by the high recovery of target cells and high purity with high-throughput characteristics. The MF consisted of a series (150) of sinusoidal channels that were 20 µm wide and 150 µm deep and demonstrated purities >90% [27,28,29], specificity of 88–99% [31], and 70–90% recoveries for epithelial cancer cells [27,29]. We would like to note that the biology of CLCs and, thus, their analysis is much different from CTCs (circulating tumor cells). For example, the size of CTCs is in the range of 10–23 µm, and CLCs can range from 6 to 12 µm, which is similar to the size of leukocytes. Thus, size-based cell isolation techniques cannot be used for enriching CLCs, because a significant number of highly abundant leukocytes would be enriched as well. In addition, the selection markers used for enrichment of the CLCs can be co-expressed on non-leukemic cells. In the case of CTCs, many times, epithelial cell adhesion molecules (EpCAMs) are used for enrichment, which are not expressed on leukocytes and also, few, if any, epithelial cells are found in circulation. In the present case, we used CD19 as the selection marker, which is a type-I transmembrane glycoprotein typically expressed on B cells throughout their different stages of differentiation. Thus, both CLCs and normal B cells will express this antigen; discrimination between normal and leukemic B cells will depend on the immunophenotyping of these cells.

Here, we demonstrate the use of this technology for the analysis of CLCs in the circulation of children diagnosed with precursor B-ALL and compared our results to conventional MRD measurements performed using MFC on BMA. We demonstrate that the microfluidic (MF)-MRD (MF-MRD) test using peripheral blood is more sensitive than MFC-MRD and could secure CLCs for further testing using molecular profiling or FISH. The use of peripheral blood for MRD testing carries important benefits, especially for pediatric patients; it is far less invasive than BMA sampling, and there is no need for anesthesia/sedation, both of which allow for more frequent MRD testing, which is important for the monitoring efficacy of treatment in acute types of diseases. Sampling of peripheral blood may be also more accurate, as blood is a homogenous medium, unlike BMA due to its non-uniform spatial composition, which can lead to sampling errors.

## 2. Results

### 2.1. Evaluation of the Processing Pipeline for the Affinity Isolation of CLCs

The rare cell selection chip [32] was made from cyclic olefin polymer (COP) via injection molding, with the chip surfaces covalently decorated with an anti-CD19 mAb (Figure 1A(a–c)) for the affinity selection of target cells. Several aspects of the sinusoidal architecture were optimized (125 µm radius of curvature, 25 µm width, and 150 µm depth) to maximize recovery, throughput, and purity of the isolated cell fractions [27,28,33]. The sinusoidal architecture generated centrifugal forces that propelled cells towards a mAb-coated channel wall with a magnitude that varies with cell diameter, density, and forward velocity [34]. In the cell selection assay, a blood sample enters the selection device through a single inlet channel, uniformly passes through a parallel array of 150 sinusoidal mAb-laden selection channels [28,34,35],and exits through a single outlet channel. The shear forces generated when processing blood are high, therefore reducing the number of WBCs that non-specifically stick to the channel wall and, consequently, provides high specificity of isolated cells but does not eliminate affinity interactions between Ag and surface-bound mAb [34]. The shear forces are similar to physiological conditions; we have demonstrated that the vast majority of cells survive such processing conditions, as deduced by observing cell viabilities, which are ~90% after chip processing [29]. After blood processing, the chip was washed and isolated cells were ready for release and enumeration or any other downstream application.

Following the release of the selected cells from the surface, the eluent was collected in an assembled cytospin funnel with a lysine-modified glass slide and was cytospun using a centrifuge, which more fully automated the sample processing when compared to our previously reported assay. We improved the processing pipeline and developed a more optimized methodology compared to our previous work as well [31], where the captured AML CLCs were immunostained on-chip, manually released into a 96-well plate and visualized/enumerated using fluorescence microscopy. This workflow required manual handling of cells and placement into the wells of the titer plate [31]. Unfortunately, the assay lacked automation, making it prone to a potential loss of cells and/or reproducibility issues due to operator-dependent CLC handling, which was directly addressed in this work.

The full protocol is provided in Appendix A. We evaluated the efficiency of the cytospin process using a SUP-B15 line as a model and as shown in Appendix A for different centrifugal speeds and processing times (Appendix A). The same volume (200 µL) of the cell suspension was (i) deposited within a flat bottom 96-well plate with lysine-modified glass and (ii) deposited into the cytospin funnel and centrifuged onto a cytospin slide. During the cytospin procedure, cells were moved from the cytospin funnel and deposited onto a microscope slide, while the buffer was absorbed by the filter positioned between the funnel and the glass slide. Cells deposited in the 96-well plate were centrifuged to force them to the bottom of the well. The supernatant was carefully removed without disturbing the cells and replaced with 3% PFA (15 min) to fix cells and attach them to the lysine-modified glass slide. Cytospin-deposited cells were also fixed and attached to the microscope slide. Cells on the glass slides and on the plate were prorated and stained with nuclear DAPI stain (Appendix A) and counted to determine efficiency. The highest transfer efficiency (82%) was demonstrated for a 5 min applied centrifugal force of 630× *g*. Stronger centrifugal forces (1120× *g*) were found to severely damage cells. Ultimately, for the processing of clinical samples, 630× *g* for 5 min was used to deposit cells onto the microscope slide.

Cells were isolated and immunostained with B-lineage-specific Abs with the aid of an automated stainer instrument (Figure 1A(d,e)). In this work, we used a microfluidic with anti-CD19 mAb attached to the channel surfaces for the isolation of cells expressing CD19 antigens (i.e., B cells and CLCs). B-ALL CLCs are classified according to the stage of differentiation in the bone marrow as pro-B, common, and pre-B [36]. Appendix A presents the phenotypes of B-ALL and their comparison to the normal mature B cells and type 4 hematogones found in blood [37,38,39]. Following blood processing, the chip was washed and isolated cells were released and deposited onto a glass slide. The following markers were useful in the classification of CLCs based on immunofluorescence: anti-Terminal deoxynucleotidyl transferase (TdT), anti-CD34, and anti-CD10 mAbs. Based on the reported data, 100% of CLCs express CD19, 91% of CLCs show expression of TdT in the nucleus, and ~89% and 76% of CLCs express CD10 and CD34, respectively [9]. TdT is an intracellular and intranuclear DNA polymerase that catalyzes the template-independent addition of deoxynucleotides to the 3′-hydroxyl terminus of oligonucleotides [40]. TdT is expressed in both T and B-precursor cells and ultimately was used in our work as a marker for the identification of CLCs.

Glass slides with stained cells were imaged using an epifluorescence microscope. Constructed circular (d = 28 mm^2^) fluorescence images are shown in Figure 1A(f), as a composite of 100 images. The images were collected using a 10× objective with a total of 36 min acquisition time for four colors: 20 ms DAPI, 600 ms FITC, 1500 ms Cy3, and 2000 ms Cy5 to phenotype the cells deposited onto a glass slide. Using this methodology, we were able to analyze pediatric B-ALL patient blood samples. The sample processing workflow is summarized in Figure 1B.

We evaluated the fixation efficiency of the selected cells to ensure that they were not lost during the staining process using an autostainer. Glass slides with DAPI-stained cells attached to a lysine-modified surface were mounted into the autostainer, and the optimized protocol for cell staining and washing was executed (see the Appendix A for the staining protocol using the autostainer). Cells were enumerated immediately after attachment to the glass slide by DAPI staining and again counted after executing a mock multi-step staining protocol with PBS/0.005% Tween and counts compared (Appendix A). There were, on average, 5.3 ± 4.0% more cells detected after the washing step due to lower background and better single cell signal. Before washing, multiple cells positioned close to each other were counted by the automatic analysis system as a single event, while, after executing the washing step, these cells were counted as individual events. Importantly, following the automatic staining, we observed no cell loss on the slides. The presented semi-automated assay was used for the analysis training sets with B-ALL model samples and clinical samples.

### 2.2. Choice of Antibodies for the Selection and Staining of CLCs

For the evaluation of the assay, we used model CLCs, which consisted of a SUP-B15 (ATCC CRL-1929™) cell line. SUP-B15 is a B-lymphoblast cell line isolated from the marrow of a White, 8-year-old, male patient. These cells express CD19 antigens, as verified via flow cytometry and immunophenotyping (Appendix A, see protocol in the Appendix A) with anti 4G7-2E3R mAb. This cancer cell line’s antigen expression showed a ~10× larger number of CD19 receptors when compared to normal B cells (~150,000 vs. 15,000 receptors, respectively), which can result in the more efficient isolation of CLCs than normal B cells owing to higher cell surface density of the receptors used for the isolation of CLCs in this case [41].

The model cell line showed positive for CD19 antigen, TdT, and CD10 receptors but was negative for CD34 (Figure 2B,C). Different clones of anti-CD19 antibodies were evaluated for the selection of CLCs (Figure 2D,E). Among the anti-CD19 mAbs tested, it appeared that CD19 antigens on the SUP-B15 surface showed the highest fluorescence intensity after incubation with clone 4G7-2E3R, as shown by our flow cytometry data (Figure 2E). This clone was used for the affinity isolation of CLCs from clinical samples. The staining of the CLCs with TdT showed a “granular pattern” distributed in the nucleus (Figure 2C), while normal B cells or precursor B cells (i.e., hematogones) showed distinctly different staining localized within the cytoplasm or nuclear membrane (Appendix A). While enumerating CLCs, it was important to distinguish hematogones from lymphoblasts (i.e., CLCs), and the nuclear pattern of immunofluorescence using mAbs targeting TdT was shown as an effective method for distinguishing these types of cells using microscopy [42]. The use of TdT in MFC has been reported; however, owing to the difficulty in localization of the marker in the cell using MFC, it is much more difficult to distinguish leukemic blasts from hematogones.

The efficiency of the SUP-B15 cell line recovery was evaluated by microfluidic cell mass balance [28]. The efficiency of the cell affinity isolation was flow rate-dependent; it increased with the increasing linear flow rate up to 2 mm/s (25 µL/min). The recovery of SUP-B15 cells spiked into healthy donor blood was 26 ± 9% (n = 3) at 45 µL/min and 48 ± 7% (n = 3) for 75 µL/min (2 mm/s). As a reference, only 3 ± 2% (n = 3) of the model cell line were found non-specifically captured on a pristine chip (no UV/ozone activation and no antibody attachment). Cells following isolation were released from the chip by means of enzymatic cleavage of the uridine-containing oligonucleotide linker with the antibody attached to the 3′ end [29]. The release efficiency was tested with the model cell line and clinical samples. The release of the cells bound to the chip surface via the antigen–antibody interaction is specific and should yield only CD19(+) cells with minimal background. In healthy blood, the release of CD19-bound SUP-B15 cells was 87 ± 11% (n = 6) using USER™ enzyme incubated for 40 min at 37 °C.

### 2.3. B-ALL Pediatric Patients

We enrolled 20 B-ALL pediatric patients in a longitudinal study for monitoring MRD using CD19-expressing CLCs. Patient characteristics and disease hallmarks are shown in Table 2 and Appendix A, respectively. The patient cohort was composed of 55% males. Most (85%) of the patients were between 1 and 10 y, and 15% were between 10 and 19 y. At the time of diagnosis, 90% of the patients showed very high (>50 × 10^6^/mL) white blood cells (WBCs). None of the patients had CNS involvement. Sixty percent of patients’ disease involved the presence of chromosomal aberrations in B-ALL blasts. Factors such as age, WBC count, and chromosomal abnormalities were used for risk stratification of B-ALL patients at the time of diagnosis. Initially, 15 patients’ B-ALL was classified as standard-risk (NCI, pt # 2, 3, 4, 5, 6, 7, 8, 12, 14, 16, 17, 18, 19, 20, and 21); however, following evaluation of the therapeutic response (i.e., MRD), this classification was changed in pt # 6, 7, 20, and 21; their disease classification was changed to high risk.

As a standard of care in the hospital, there were two MFC-MRD tests performed (day 8 from blood and 29 of treatment from BMA; see the testing schema in Figure 3A). Only for three patients (#8, #9, and #21), MFC-MRD from BMA was also tested on day 85 of treatment [43].

### 2.4. Treatment for B-ALL Pediatric Patients

Patients received induction therapy, which consisted of a combination of drugs: cytarabine, vincristine, dexamethasone, pegaspargase, and intrathecal methotrexate, and were given them over 4 weeks. Patients then received consolidation (intensification) with multiagent therapy, including vincristine, mercaptopurine, and methotrexate. During maintenance therapy, the drugs administered included mercaptopurine, methotrexate, steroids, and vincristine; intrathecal methotrexate was administered throughout the course of treatment [44,45]. The treatment regimen adopted for patients with ALL is often determined by the leukemia’s Philadelphia chromosome status and the age of the patient. Patients with Philadelphia chromosome-positive (Ph+) B-ALL receive a tyrosine kinase inhibitor in combination with chemotherapy. No patients in our study were identified with Philadelphia chromosome.

Patients’ complete remission was defined as <5% blasts in the bone marrow, no evidence of extramedullary disease, and recovery of peripheral blood counts with an absolute neutrophil count ≥1000/µL and platelet count ≥100 × 10^9^/L. Complete remission with incomplete recovery was defined as <5% blasts in the bone marrow, no evidence of extramedullary disease, and residual neutropenia, i.e., an absolute neutrophil count <1000/µL and/or platelet count <100 × 10^9^/µL. Negative minimal residual disease (MRD) was defined as <0.01% leukemia detected in a bone marrow aspirate specimen by multiparameter flow cytometry (MFC) [36,46].

### 2.5. Improved Sensitivity of MF-MRD for Rare Cell Analysis Compared to MFC-MRD

Pediatric patients enrolled in the study were diagnosed at the Children’s Mercy Hospital, Kansas City, MO, USA. Blood samples were collected on days 8, 15, 22, 29, 57, and 85 during induction and consolidation treatment (Figure 3A). CLCs were enumerated as cells that expressed CD19 (enrichment antigen), showed the presence of TdT in the nucleus, and with detectable expression of CD10 or CD34. Identification of CLCs using immunophenotyping allowed for the differentiation of CD19(+) mature B cells (i.e., normal cells) from immature CD19(+) CLCs. A summary of the data secured in this study is shown in Figure 3 and Appendix A.

The MF-MRD was measured by calculating the % of CLCs (Figure 3B) with respect to mononuclear cells that were enumerated during the standard-of-care CBC test performed in the hospital and on the same day that CLC enumeration was performed. By expressing the MF-MRD results in this way, the values could be compared to the standard of care (MFC-MRD either measured from blood or BMA). The LOD for the MFC-MRD is marked by the green dashed line and was found to be 0.01% (Figure 3B).

To establish the LOD for the MF-MRD, we tested healthy donors’ blood using the same protocol for testing B-ALL patient’s blood (Appendix A). The average nucleated cells that were TdT(+) was 8.8 ± 5.4/mL. The threshold was established as the average number plus 3 × SD of detected TdT(+) cells in healthy donor blood, which was 25/mL of blood containing ~5 × 10^6^ mononuclear WBCs. Therefore, the LOD was established as 25 cells/5 × 10^6^ cells (i.e., 5 × 10^−4^%). We concluded that, above the 5 × 10^−4^% threshold, MRD was considered positive (Appendix A).

In 41 paired MRD tests for 20 patients using MFC and MF, MFC MRD(+) was detected in 14/41 tests, while 27/41 tests were MRD(−). For the MF-MRD assay, MRD(+) was detected in 22/41 tests, while 19/41 tests were MRD(−). The consensus between the MFC and MF-MRD tests was seen in 19 cases (12 MRD(−) and 7 MRD(+)). MFC did not detect MRD in 15/41 tests where MF detected MRD. In 7/41 tests, MFC detected MRD, while MF did not (Appendix A). Overall, out of 116 MF-MRD tests performed, 57% (66) showed MRD(+), while, for MFC-MRD, 34% (14/41) MRD tests were positive.

Higher average CLC counts were observed for high-risk B-ALL patients compared to standard risk patients (Figure 3B,C and Appendix A), indicating the ability of the MF-MRD to differentiate the disease burden. The average CLC counts detected during treatment (Figure 3C) were compared between standard and high-risk patients. Statistically significant differences between CLC counts were seen on day 57 of treatment (i.e., consolidation therapy (Figure 3C)). The burden was 23 CLC/mL vs. 124 CLC/mL for standard and high-risk patients, respectively.

In patient #10 on day 57 of treatment, MF-MRD detected a large burden of disease (377 CLC/mL). This patient died of disease before day 85 of treatment. For two patients (#1 and #4), MF-MRD was positive, but we cannot attest to the patients’ disease status, because they relocated and were no longer enrolled in the study. For 14 patients, the last day of consolidation therapy showed that all were MF-MRD-negative and at their last physical checkup, all were in remission (NPV = 100%). For three patients (#8, #9, and #19) on day 85, MF-MRD was positive, but these patients were considered in remission ~4 y post-diagnosis (FP = 17.6%). We should note that patient #19 had a very low level of positive MRD (25.5 CLCs/mL with a threshold of 25 CLCs/mL). We built a receiver operating curve for all 116 data collected and calculated the area under the curve, which was found to be 83% (specificity 85.7% and sensitivity 71%).

During MF-MRD testing over the course of induction and consolidation therapy, the phenotypes of the CLCs varied (Figure 3D). We identified CD19(+)/TdT(+) CLCs with the following phenotypes: CD34(+)/CD10(+),CD34(−)/CD10(+), CD34(+)/CD10(−), and CD34(−)/CD10(−). During treatment on days 22 and 85 in high-risk patients, CLCs that were TdT(+)/CD34(+)/CD10(−) were no longer detected, but the percentage of TdT(+)/CD34(+)/CD10(+) expanded on day 85. In these patients, the dominant population were CLCs lacking CD34 but positive for CD10 (Figure 3E). The phenotype of the blasts can be used for B-ALL clone assessment into different subgroups (Appendix A). For example, the expression of CD34 and TdT indicates immaturity and is a characteristic of pre-B-ALL (Appendix A).

### 2.6. Fluorescence In Situ Hybridization (FISH) of CLCs

FISH for pediatric B-ALL uses a panel of probes to search for relevant aberrations typically found in B-ALL blasts. During follow-up testing, the entire FISH panel is not used, just probes to observe evidence of a therapy response. In the cohort of our B-ALL patients, 60% (12/20) were identified with leukemic cells having chromosomal aberrations. In 35% of patients participating in the study, a double trisomy (+4,+10) was identified, and an additional 25% had blasts that carried a t(12;21) aberration. Usually, ~30% of B-ALL patients’ leukemic cells had hyperdiploidy, 25% demonstrated t(12;21), and 5% and 10% patients’ blasts had t(9;22) and MLL translocations, respectively. Once aberrations were detected, this information can aid in determining the proper treatment regimen. For example, the detection of t(9;22)(q34;q11.2), the so-called Philadelphia chromosome, is used to assign B-ALL patients to a specific targeted therapy [47].

Selected CLCs were interrogated by FISH using probes for aberrations identified at the time of diagnosis. As a standard, 200 cells’ nuclei were examined. Images of nuclei with aberrant chromosomes or abnormal configuration are presented in Figure 4A. A double trisomy (+4,+10) was tested with Cytocell D4Z1 (4p11.1q11.1) and D10Z1 (10p11/1q11.1) probes (Chromosome 4 and 10 are red and green, respectively), while the t(12;21)(p13;q22) aberration was tested with Cytocell TEL/ETV6 (12p13) and RUNX1 (21q22) probes (i.e., ETV6 is the red probe, and RUNX1 is green). If the cell was found to be positive for trisomy, three chromosomes for both 4 and 10 were observed. For t(12;21)(p13;q22), two occurrences of red/green probes that overlap per cell were seen. For t(12;21)(p13;q22), two occurrences of red and green probes were seen that overlapped in the nucleus (Figure 4A).

The abnormalities in the nuclei detected by FISH are listed in Figure 4B. For example, in blasts isolated from the blood of pt #3 on day 29 of treatment, we detected five nuclei with two copies of chromosome 4 and three copies of chromosome 10 and two nuclei with one copy of chromosome 4 and two copies of chromosome 10. These cells do not appear to be normal; however, they did not display a typical pattern for the trisomy 4/10 (three chromosomes for both 4 and 10). Interestingly, in blasts isolated from patient #5 on day 85, we detected abnormal cells with one red/green overlapping signal and one signal for ETV6 and RUNX1 (red and green, respectively). One cell had a nucleus that contained two ETV6 red signals and three RUNX1 green signals. A clinical sample collected from a patient diagnosed with B-ALL showed cells having additional chromosomes 4 or 10; it is possible to have cells with a pattern other than a typical fusion, if, for example, the clonal evolution of leukemia cells took place or if the subclones existed at the time of diagnosis but were not detected. There also may be random cells with an alternate pattern of chromosomal aberrations. The significance of these observations is yet to be determined.

### 2.7. Molecular Profiling of CLCs

We evaluated the expression of the genes reported to have predictive or prognostic values in B-ALL patients [48,49]. Our hypothesis was that, if the genes are the signature of B-ALL leukemia cells, we should be able to demonstrate the differential expression of these genes in the isolated CLCs during longitudinal testing. The expression of the following mRNA transcripts: *CD19*, *WNT5A*, *CCND2*, *IL2RA*, *SORT1*, *FLT3*, and *DEFA1* was evaluated [50,51,52,53,54,55,56]. The information in Appendix A describes the function of these genes in B-ALL, and Appendix A provides the sequences of the primers as reported [48].

Expression levels of the listed mRNA transcripts were obtained from mRNA extracted from the affinity isolated B cells from healthy donors (HD 1–4), and clinical samples of RT-ddPCR were used to secure the expression data. The listed transcripts in HD showed appreciable expression of *CD19* mRNA (113 copies/ng total RNA). Other transcript abundance varied between 2 and 50 copies/ng of total RNA. For comparison, the expression of *CD19* mRNA in the SUP-B15 B-ALL cell line was 200× higher than in HDs with 22,500 copies/ng total RNA, which agrees with our flow cytometry data. Similarly, *CCND2*, *FLT3*, and *SORT1* mRNAs were much more abundant in the SUP-B15 cells than in normal B cells isolated from HD (Figure 5A). Negative RT controls show no appreciable expression. For illustration, results for RT(−) and RT(+) ddPCR performed on total RNA isolated from affinity selected CD19(+) cells (normal and CLCs) for pt #15 on day 8 of treatment are shown in Figure 5B. Microfluidic isolated CD19(+) cells (normal B cells and CLCs) from four patients (#15 (high-risk, HR), 16 (standard-risk, SR), 17(SR), and 18(SR)) were interrogated for gene expression on day 8 of treatment (Figure 5C). The profiles differed significantly from those secured from HDs. In all patients, except for pt #17, *CD19* mRNA copies were more abundant. Other genes (i.e., *CCND2*, *DEFA1*, *FLT3*, *IL2RA*, *SORT1*, and *WNT5A*) were either slightly more abundant or on par with the levels detected in HDs. For this patient on day 8, ~30% of the CD19(+) cell population consisted of CLCs (on par with LOD at 25 CLCs/mL) and patient MF-MRD was deemed negative. For pt #15, 16, and 18, *CCND2* mRNA was highly abundant (3000 copies/ng total RNA), *DEFA1* was overexpressed in pt #16 and 18, and *FLT3* and *IL2RA* were elevated in pt #15 and 16 (Figure 5C).

For pt #15 (HR), we evaluated gene expression longitudinally (day 8–day 29, which corresponds to induction therapy, Figure 5D). Profiles differed for different days over the course of treatment, with one very prominent observation of the continuous loss of CD19 mRNA expression between days 8 and 29. Herein, we probed the *CD19* mRNA spanning exons 8–9 (encoding the cytoplasmic domain of the CD19 domain, not extracellular), and it appeared that, during the induction therapy, this CD19 isoform decreased significantly when compared to day 8 of treatment. The only abundant gene on day 29 was DEFA1. The MF-MRD was negative on day 29 for this patient. The same trend was observed for patient #16 (SR), in which *CD19* mRNA expression was lower than in HDs, and the only appreciable expression was from *DEFA1* on day 22 of treatment. For pt #16, the MF-MRD was positive on days 8 and 15 but negative on day 22. In both patients, the MF-MRD(−) status coincided with silent *CCND2*, *FLT3*, *IL2RA*, *SORT1*, and *WNT5A* but higher copies of *DEFA1* (Figure 5E).

The expression changes in *CD19* mRNA when probed with primers spanning exon 8–9 (cytosolic domain) could indicate the potential downregulation of CD19 protein, which is used for the affinity isolation of CLCs. Thus, we evaluated the expression of mRNA responsible for translating the extracellular part of the CD19 protein and alternatively spliced *CD19* mRNA isoforms in the extracellular domain. Alternative splicing of mRNA results in multiple transcript variants encoding distinct isoforms of CD19. Figure 5F illustrates isoforms of *CD19* mRNA, including skipping of exon 2 (ΔEx2), the partial deletion of exon 2 (ΔEx2part), and deletion of exons 5 and 6 (ΔEx5–6). The intact exons 1–4 of *CD19* mRNA (NM_001178098) are responsible for translating the extracellular part of the CD19 full-length protein, while the transmembrane domain is encoded by exons 5 and 6 (Figure 5F). The cytosolic part of CD19 is encoded by exons 7–15. Using a set of primers spanning different exons in *CD19* mRNA and performing end point PCR with gel electrophoresis to determine the amplicon size, we could deduce the presence of alternatively spliced variants of *CD19* mRNA. The primers were designed by Sotillo et al. [57] (Appendix A).

Following the gel electrophoresis, we identified (see Figure 5F) alternatively spliced *CD19* mRNA, including transcripts encoding for the full CD19 protein that is used for the affinity isolation. When mRNA spanning exons 4 and 8 were reverse-transcribed and amplified, in all samples tested, including HD, the presence of an amplicon of 640 bp indicated a mRNA variant that encoded the full-length protein. In the control total RNA from the SUP-B15 cell line, in addition to full-length mRNA, a shorter 331 bp long amplicon was detected, indicating missing exons 5 and 6 (ΔEx5–6) in CD19 mRNA (Figure 5G). ΔEx5–6 in *CD19* mRNA is responsible for the slower growth of cells in culture [57]. As a negative control, RNA was extracted from the Molt3 cell line (T-ALL model), and as expected, no *CD19* mRNA transcripts were detected, as CD19 is exclusively expressed in B-cell lymphocytes or CLCs.

To evaluate potential variants with missing exon 2, mRNA spanning exon 1 and 4 and 1 and 5 were reverse-transcribed and amplified (Figure 5H,I). Based on the product detected following end point PCR, we concluded the presence of full mRNA transcripts (640 bp and 800 bp for the exons 1–4 range and 1–5 range, respectively) and also variants with ΔEx2 (374 bp and 533 bp for exons 1–4 and 1–5, respectively) and ΔEx2part (509 bp and 669 bp for exons 1–4 and 1–5, respectively), indicative of the presence of the truncated extracellular domain of CD19. For pt #16 on days 8 and 15 of treatment, we detected CD19(+) cells’ mRNA variant coding for full protein, ΔEx2 part, and ΔEx2. On day 22, we did not detect extracellular coding mRNA (however, the patient was MF-MRD(−)). For pt #14 and #15, isolated CD19(+) cells had both *CD19* mRNA variants (full-length mRNA and ΔEx2). Among our patient samples, only pt #17 CD19 mRNA showed exclusively the full-length mRNA variant. It is important to note that alternative splicing is not exclusively observed in B-ALL patients. In HD (1–3), different *CD19* mRNA alternatively spliced variants were detected (Figure 5G–I). Because these variants were found in affinity isolated cells, targeting the full-length extracellular domain would imply that the full-length and truncated CD19 protein coexist on the surface of the cell.

## 3. Discussion

Despite significant improvement over the past 50 y in the overall survival of children diagnosed with B-ALL, 10% of patients will experience a relapse and die from disease [58]. One critical way to improve survival rates in B-ALL is to identify patients that have a high risk of relapse based on MRD and treat these patients with an intensified chemotherapy or potentially offer an allogeneic hematopoietic stem cell transplant. For B-ALL pediatric patients, the rate of response to chemotherapy is a strong prognostic factor. The kinetics of the treatment response is measured by MRD detected in the bone marrow at predefined timepoints to identify patients at high risk for relapse. It is important to note that MRD negativity is not defined as the absolute absence of residual leukemia cells but rather lack of leukemic cells at a predefined threshold value that is based on the analytical figures of merit of the measurement system. The LOD of any MRD assay determines the threshold level below or above which the patient’s disease is deemed MRD(−) or MRD(+), respectively. Therefore, a better definition of MRD should be “measurable” rather than “minimal”, as postulated by Brüggemann [4].

We performed a paired comparison of the MF-MRD test from blood with the standard-of-care MFC-MRD assay detected from BMA. Unprocessed blood samples for the MF-MRD test were analyzed within 6 h following blood draw. In contrast, MFC-MRD based on testing of the BMA is not as simple. Aside from the fact that the BMA collection process is highly invasive, especially for pediatric patients who are sedated during the procedure [59], securing the appropriate quality sample of the BMA requires skilled personnel and extensive sample preparation prior to the analysis. Poorly obtained BMA can be diluted (i.e., hemodilution), or owing to the non-homogeneous nature of bone marrow, sampling errors is intrinsically associated with this procedure, which can lead to false-negative results. In pediatric patients, the presence of hematogones in bone marrow can lead to false MRD as well [37,59,60].

The processing of the BMA sample for MRD is also not straightforward, as the MFC-MRD protocols require red blood cell lysis before staining of nucleated cells. This step can be associated with WBCs/leukemic cell losses. For the detection of rare cells, statistics dictate that the assay must interrogate a vast number of cells, as the detection sensitivity depends on it, while, at the time of disease diagnosis, testing a relatively small number of nucleated cells is enough to detect a high concentration of leukemic blasts that can comprise 90% of all nucleated cells. However, during longitudinal testing throughout treatment, at least 1 × 10^6^ cells must be evaluated to statistically account for the presence of rare leukemic cells [59]. MFC-MRD is deemed (+) when leukemia cells are ≥0.01% of the mononucleated cells [60]. Typically, a three to four-color MFC reaches a detection sensitivity of 0.01% by interrogating 5 × 10^5^ cells (i.e., detects 50 aberrant cells). To reach a sensitivity of 0.001%, it would thus be required to interrogate ~5 × 10^6^ cells. Therefore, cell losses can be potentially detrimental to the assay’s LODs. New generation 10-color MFC demonstrated a sensitivity of 0.0002% [61] by interrogating 4 × 10^6^ BMA cells (at ~4000 cells/s) that allowed for the detection of leukemia cells in samples reported to be MRD(−) using standard MFC-MRD. However, immunostaining with 10 markers required special dyes and sophisticated instruments and is not trivial, limiting its broad range applicability. A seemingly prosaic but important challenge may be the fact that data analysis required vast computational power because of the massive size of the raw data files obtained when millions of cells are interrogated. While new generation MFCs are being developed, these are not standard of care and do not address the limitations of the cellularity limits of BMA, particularly in aplastic BMA, with the heterogeneity of the bone marrow space prone to sampling errors, as noted earlier [62].

The important question arises: can leukemia cells (i.e., CLCs) be found in the circulation at levels that allow for MRD testing during all phases of treatments? The work of Van der Velden et al. [63] using a PCR assay by detecting immunoglobulin (Ig)/T-cell receptor (TCR) gene rearrangements reported for B-ALL that the burden of leukemia cells found in BMA was up to 1000-fold higher than in the circulation. This would be a significant challenge for a conventional MFC to detect such a low burden of CLCs. However, for more sensitive methods, such as MF-MRD, detection from the circulation is feasible. It is important to note that the detection of Ig heavy chain (IGH) gene rearrangements in PCR also presents limitations, as these targets can be lost during the course of disease progression [64]. The loss of these PCR targets can lead to false-negative MRD results. Interestingly, in the same BMA vs. blood MRD study, 11% samples had MRD(+) in blood, while the sampling of the BMA was MRD(−) [63]. Unfortunately, the detection of MRD for B-ALL patients using PCR is currently limited to <50% of pediatric patients due to unidentified patient-specific markers [65].

Arguments for the feasibility of detecting the presence of CLCs in circulation were reported with NGS. The sensitivity and specificity of NGS-MRD calculated for a blood TCR target were 94% and 93%, respectively, and were actually higher than the BMA TCR MRD (92% and 83%, respectively) [66]. In a work by Wood et al., NGS identified ~39% more MRD positive patients compared to MFC-MRD. Even at the 0.01% MFC-MRD threshold, NGS identified MRD in more patients than MFC [19]. While NGS is a powerful method for MRD detection, it is not without challenges, such as securing enough gDNA from B-lineage cells for testing, quantification of the B-ALL-related sequences for MRD level determination, and the requirement for gDNA sequencing before treatment initiation to identify sequences indicative of leukemic clones needed for the follow-up MRD testing. The MF-MRD assay presented herein uses a peripheral blood sample that can be implemented for frequent MRD testing. Lineage-associated antigen CD19 was used for the isolation of B cells and B-ALL cells, while maturational stage-associated antigens such as CD34, CD10, and TdT were used for CLC identification [67]. Affinity isolated cells were released from the capture surface and thus made available for any downstream application (i.e., enumeration, sequencing, mutation screening, and FISH). All of these applications were demonstrated in this report and from our past report [30]. In our previous works, we presented the effects of physical dynamics and device architecture on the efficiency of affinity selection of cells in sinusoidal microchannels [27,29,31,34,68]. The process of affinity-selection of cells using a sinusoidal chip consists of two phases: (i) initiation of contact between a cell and the mAb bound to the surface and (ii) binding of the rolling cell with surface-bound mAbs. With an increasing linear velocity in the curvilinear channel, the resulting centrifugal forces increase the delivery of cells to the outer channel wall. Inherently, the centrifugal forces have a lesser effect on smaller objects, including cells. For example, the centrifugal forces propel an 8 µm B cell with a velocity four times slower than they would a 16 µm diameter cell, and as a result, smaller cells do not reach the antibody decorated outer channel wall as efficiently as large cells to allow for cell surface antigen interactions with mAb. While the sinusoidal architecture of channels with a 125 µm radius of curvature, 25 µm channel width, and 150 µm depth provided high recovery (~90%) and high-throughput for cells with diameters 15–20 µm such as CTCs, the recovery of smaller cells as reported herein was 48%. Theoretically, the cell recovery could be improved by increasing the linear velocity above the currently used 2 mm/s to increase the centrifugal forces. Unfortunately, this action would shorten the residence time of the cell at the surface-bound mAb and therefore affect the overall binding kinetics of antigen and antibody binding, resulting in lower cell recovery. To increase the efficiency of the cell recovery, the channel’s width could theoretically be decreased as well; however, the consequences of such action would be disadvantageous to the purity, specificity, and throughput of the microfluidic assay. The purity of isolated rare cells would be compromised owing to the presence of larger cells in the blood (i.e., granulocytes) that would be physically trapped in channels. Owing to the narrow sinusoidal channels and fast linear velocity of processing, the fluid shear stress for blood generated in the sinusoidal channels is 13.3 dynes/cm^2^. Such high shear forces prevent “permanent” non-specific interactions with the channel walls, resulting in highly specific affinity isolation.

To streamline the MF-MRD assay, a semi-automated processing pipeline was conceived and optimized, which included deposition of the released cells onto a polylysine-treated glass slide using a cytocentrifuge, fixation/attachment of the deposited cells onto the slide using PFA, and immunostaining with the aid of an automated stainer (Figure 3). Stained cells were imaged using an epifluorescence microscope, and the developed software aided in the identification and enumeration of CLCs specifically.

We optimized the aforementioned processing steps using a B-ALL SUP-B15 cell line for the following markers (i.e., CD19, TdT, CD34, and CD10), of which CD19/TdT were the major signatures of the leukemic cells (i.e., CLCs), Figure 4A,B. For selected patients (n = 3), we performed MF-MRD analysis in duplicate, and we observed a chip-to-chip reproducibility of ~12%, ensuring the accuracy of the MRD test. While enumerating CLCs by evaluation of the staining pattern, we distinguished normal precursors B cells (i.e., hematogones) from CLCs, both of which express the selection antigen CD19. Hematogones usually show staining outside the nucleus and in the cytoplasm (Appendix A) [37]. The nucleus-specific fluorescence pattern with TdT antibodies for CLCs was shown as an effective method for distinguishing hematogones from leukemic blasts using microscopy but is not frequently used in MFC as localization of TdT cannot be easily recognized [37,42].

### 3.1. Enumeration of CLCs

CLCs have clinical significance in B-ALL for disease management, like that seen for CTCs in epithelial cancers [32,69,70,71,72,73]. The number of non-aberrant cells selected using anti-CD19 antibody is much higher (up to 20,000/mL, Appendix A) than in the case of CTCs using, as an example, anti-EpCAM mAbs as the selection marker (e.g., 10 cells/mL) [28]. This is a consequence of the biology of CLCs and not inadequacies associated with the microfluidic assay. Because normal blood cells do not express epithelial markers and epithelial cells are not typically found in the circulation, the affinity assay with anti-EpCAM mAbs shows high purity of isolated EpCAM(+) CTCs. This is not the case with anti-CD19 Ab, as both leukemia and normal B cells express the CD19 antigen, a hallmark of cells from the B-lineage. Hence, while CD19(+) CLCs are isolated with high specificity, the purity (ratio of CLCs/(CLCs + PBMCs) depends on the number of co-isolated normal B cells.

The current standard of care for the detection of MRD for B-ALL patients include MFC testing of BMA at diagnosis, the end of induction (day 29), and at the end of the consolidation therapy (day 85). Only early in treatment (i.e., day 8 of induction), MFC-MRD from blood can be performed. The presence of CLCs in blood (i.e., MRD(+)) at any of these timepoints is an indicator of ineffective therapy or chemoresistance. Usually, higher levels of MRD are associated with an increased relapse rate and poor overall survival. For example, patients with MRD < 10^−4^ following induction therapy have superior disease-free survival compared to intermediate levels of leukemic blasts ranging from 10^−4^ to 10^−3^ [47]. The benefit is pinpointing when a patient’s MRD is indicative of a relapse and can provide an earlier intervention with a potentially better outcome [74]. By applying the MF-MRD, we could detect CLCs in pediatric B-ALL patients with higher sensitivity than MFC, which could assist in guiding therapy to enable precision medicine with earlier interventions to improve the patient outcome [63].

In our MF-MRD assay, the processing of 2 mL of blood allowed for statistics to reach a LOD of 10^−5^ (0.001%) with the LOD of the MF-MRD assay at 25 cells/5 × 10^6^ cells (i.e., 5 × 10^−4^%). We concluded that, above the 5 × 10^−4^% threshold, MRD was considered positive (Appendix A). Statistically significant differences between CLC counts in SR and HR patients were found on day 57 during consolidation therapy (Figure 3B). The burden was 23 CLC/mL vs. 124 CLC/mL for SR and HR patients, respectively. During longitudinal testing in this pilot study with 20 patients, we performed 116 MF-MRD tests, while only 41 MFC-MRD were performed over the same time period. Overall, out of 116 MF-MRD tests performed, 57% (66) showed MRD(+) as defined by exceeding the threshold of 0.005%. Above the threshold of 0.01% for MFC-MRD, 34% (14/41) MRD tests were positive. The summary of the characteristics of the MF-MRD assay is presented in Table 3.

### 3.2. CLC Immunophenotype Modulation

During longitudinal MRD assessments in pediatric B-ALL patients, we observed that isolated CLCs (CD19(+)/TdT(+)) showed different immunophenotypes based on their CD10/CD34 status. For example, CD34(+)/CD10(−) CLCs in HR B-ALL patients consisted of 25% of the CLC population isolated on day 15 of therapy, then decreased to virtually none on day 22, present again on day 29 (end of induction therapy) at 50% of the isolated CLC population, and then not detected on day 85 (end of consolidation). On day 85, the dominating phenotype of the CLCs in HR patients was CD19(+)/TdT(+)/CD34(−)/CD10(+) (Figure 3E). The reduction in CD34 expression is the result of the patient’s treatment with steroids that affects cell maturation [75,76]. CD34 (and CD10) antigens can be restored when corticoid steroids are stopped [77]. While we did not evaluate phenotype changes and their prognostic values, others have, and no apparent consensus was reached. Some reported that CD34 expression in B-ALL is a predictor of poor induction therapy response because of the stemness-associated character of the CD34 antigen and possibly leukemic cell survival [78]. Another study concluded that immunophenotypic shifts were not a prognostic factor [75], while others reported that B-ALL cells with a CD10+/CD34+ phenotype were associated with favorable outcomes in pediatric patients [79].

The modulation of these antigens during treatment can be problematic for MFC-MRD, as the MFC detection mode “Leukemia-Associated Immuno Phenotype” (LAIP) relies on stable phenotypes identified at diagnosis by comparing the antigen abundance for cancer and normal cells. If the immunophenotype profile identified for the LAIP is no longer valid during treatment, the leukemia cells may not be counted. Phenotype change can also affect another MFC-MRD mode, “Different from Normal” (DFN). Therapy with steroids can affect both B-ALL and normal cells. Hence, the DFN MFC-MRD detection mode can be compromised as well. In contrast, the MF-MRD, owing to the fact that it records all phenotypes found in the CD19(+) and TdT(+) population, can follow the modulation pattern of CLCs.

### 3.3. CLCs and Chromosomal Aberrations

Our work demonstrated that CLCs sourced from blood can be used for MRD testing and the prognostication of MRD by looking for chromosomal aberrations [80]. We demonstrated herein FISH from CLCs for patients at the end of induction and consolidation therapy (i.e., days 29 and 85). The detection of specific chromosome aberrations and fusion genes is important, as it can be used to assign B-ALL patients to targeted therapy, for example, BCR-ABL1 t(9;22)(q34;q11.2) is amenable to tyrosine kinase inhibitors [23] or can be informative of a good prognosis (i.e., ETV6-RUNX1 t(12;21)(p13;q22)) [80]. Performing FISH on affinity-isolated CLCs offers the advantage of presenting to the cytogenetic laboratory preconcentrated cells with a CD19(+) phenotype. As the current standards for B-ALL diagnosis integrates immunophenotyping to identify leukemia cells (i.e., CLCs) with cytogenetics, as detailed in the 2008 WHO classification of lymphoid neoplasms [81], the use of MF-MRD is convenient, as it allows for testing of both from blood.

### 3.4. CLCs for Molecular Profiling

Based on the work of Garza-Veloz et al. [48], who identified mRNA transcripts, such as *IL2RA*, *SORT1*, *DEFA1*, and *FLT3*, as associated with a more aggressive character of leukemic cells, we performed gene expression analysis on isolated CLCs. Using mRNA from CLCs isolated in a longitudinal study, we analyzed expressions of *WNT5A, CCND2, IL2RA*, *SORT1*, *FLT3*, and *DEFA1* using ddPCR. ddPCR has advantages over qPCR, showing the ability to detect low copies of transcripts and, importantly, no need for finding stable housekeeping genes for normalization. For the tested samples, we observed that the abundance of these transcripts decreased substantially between day 8 of induction therapy to near the end or at the end of induction. On day 22 or 29, these transcripts were almost undetectable. The use of CLCs for performing molecular testing, either for a small pool of transcripts or whole transcriptome sequencing, are both feasible options. In addition to MRD detection, CLC molecular profiling can help with risk stratification. This is a notable finding, as molecular profiling could be used for prognostication in ~25% of patients who are not diagnosed with chromosomal aberrations [82].

### 3.5. Consequences of Alternative Splicing of CD19

We demonstrated the ability of testing *CD19* mRNA alternative splicing using total RNA isolated from CLCs. The recognition of different isoforms of the CD19 protein is important, as B-ALL new therapeutic approaches rely on CAR-T cells designed for the CD19 antigen [83]. Alternatively spliced CD19 mRNA, along with a missense CD19 mutation or hemizygous deletions spanning the CD19 locus [84], were reported as the reason behind the absence of a full-length CD19 protein, as shown in Figure 5F. Downregulation of the CD19 antigen in B-ALL leukemic cells may be a signature of immunotherapy-resistant cells [85]. The assay can identify patients for which different isoforms of CD19 protein must be considered for CAR-T preparation.

## 4. Materials and Methods

### 4.1. Microfluidic Chip Assembly and Capture Antibody Conjugation

The CLC selection chip consisted of a sinusoidal microfluidic device evaluated previously [28,30,31,32] that works on the principle of positive-affinity selection, where monoclonal antibodies (mAbs) are bound to the surface of the microfluidic device using a cleavable nucleotide linker containing uridine residue (Integrated DNA Technologies, Clarville, IA, USA) and specifically select target antigen-bearing cells. In this study, we employ an anti-CD19 mAb (Recombinant Monoclonal Mouse IgG1 Clone # 4G7-2E3R from R&D Systems, Minneapolis, MN, USA) for B-lineage ALL CLC isolation.

UV/O_3_ activated native cyclic olefin polymer (COP) sinusoidal microfluidic chips (BioFluidica, San Diego, CA, USA, part# 2000602) were modified according to the following protocol. For CD19 mAb immobilization, EDC/NHS (1-ethyl-3-[3-dimethylaminopropyl] carbodiimide hydrochloride/N-hydroxysuccinimide) chemistry in MES buffer (2-(4-morpholino)-ethane sulfonic acid, pH 4.8) (Thermo Scientific, Waltham, MA, USA) was used to attach a custom oligonucleotide linker, which contained a 5′-amino and 3′-disulfide modification with an internal dU residue (5′-NH2-C12-T8CCC TTC CTC ACT TCC CTT T-U-T9-C3-SS-C3OH, Integrated DNA Technologies) to the surface [29]. Zeba spin desalting columns (7K MWCO) ((Thermo Scientific) were used to purify the CD19 capture antibody (Recombinant Monoclonal Mouse IgG1 Clone # 4G7-2E3R, R&D Systems, Minneapolis, MN, USA) before attaching to the bifunctional linker to the surface of the chip. DTT (DL-1,4-dithithreitol, molecular biology grade, Acros Organics, Geel, Belgium) and sulfo-SMCC (sulfosuccinimidyl-4-(N-maleimidomethyl) cyclohexane-1-carboxylate) (Thermo Scientific) were used for the oligonucleotide–antibody attachment chemistry. After the antibody was added, a protein-stabilizing cocktail (Thermo Scientific) was added to each chip, and the chips were placed in a refrigerator (4 °C) before use. The detailed procedure reported in [29] is also presented in the Supporting Materials.

### 4.2. Clinical Samples

Healthy donor blood samples were obtained from the Biospecimen Repository Core Facility at the University of Kansas Medical Center (KUMC) under an approved IRB. All pediatric patients were treated at Children’s Mercy Hospital in Kansas City, MO, USA. Patients with a pathological diagnosis of precursor B-cell ALL, ages 1 to 18 years of age, were screened and enrolled prior to induction day 8 chemotherapy. For both healthy donors and B-ALL patients, peripheral blood samples (5 mL) were drawn by venipuncture into Vacuette^®^ containing EDTA (Thermo Fisher) tubes. Characteristics for the 20 patients enrolled in the study are detailed in Table 2 and Appendix A.

### 4.3. Blood Sample Collection and Processing

Samples were collected on days 8, 15, 22, 29, 57, and 85 of treatment. Flow cytometry was performed for the patients on days 0 and 29, and CBCs (complete blood counts) were performed each day of collection. FISH analysis was performed as well. Both induction and consolidation chemotherapy were personalized for the patient, depending on their risk scores.

Whole blood (2–6 mL) was collected into EDTA vacuum blood collection tubes and was processed within 6 h of collection. Before blood processing, chips were washed with 2 mL 0.5% BSA in PBS at 40 µL/min. Isolation of CLCs from patient blood was performed using 16-channel LiquidScan^®^ system (BioFluidica, part# 1000201) and an automated protocol optimized for CLC isolation reported below. The blood collection tubes were placed directly within the liquid handling robot, and a pair of pipette tips positioned within the liquid handling robot was used to introduce the blood into the chip; one pipette tip was operated in a push mode and the other in a withdrawal mode to make sure that blood was pumped through the chip at a constant volume flow rate. Two milliliters of blood was run through the chip at 75 µL/min for 40 min. Following blood introduction, the chip was rinsed with 2 mL of 0.5% BSA in PBS at 110 µL/min for 20 min to rinse away any uncaptured cells.

To release the cells from the chip, a mixture of USER™:UDG enzymes (Uracil Specific Excision Reagent, New England Biolabs, Ipswich, MA, USA, contains Uracil DNA glycosylase and DNA glycosylase-lyase Endonuclease VIII; extra UDG from Thermo Fisher was added in a 1:1 USER™:UDG ratio) was added to the chip (100 µL total:25 µL/min, 4 min), and the chip was incubated at 37 °C for 1 h. The two-enzyme system cleaves an uracil residue in the bifunctional oligonucleotide linker as previously described [29].

The cells were released onto a polylysine-treated glass slide using a cytology funnel and cytology centrifuge (Cytospin, Thermo Fisher Scientific, 1800 rpm, 7 min, high acceleration). Slides were positioned in an autostainer (Lab Vision Autostainer 360, Thermo Fisher), and the protocol for staining was executed (Appendix A). Cells were fixed to the slide with 2% paraformaldehyde (PFA) for 15 min and immunostained for surface markers with anti-CD34-Cy3 (Bioss) and anti-CD10-Alexa Fluor 647 (FAB1182R from R&D Systems and CB-CALLA-APC from eBioscience, San Diego, CA, USA) antibodies for 45 min. Next, the slides were treated with 0.1% Triton-X100 (Sigma-Aldrich, St. Louis, MO, USA), and staining for nuclear markers was performed with anti-TdT-FITC antibody (Clone E17-1519 from BD and MHTDT01 from Invitrogen, Waltham, MA, USA) or DAPI (4′,6-diamidino-2-phenylindole, eBioscience).

### 4.4. Cell Culture

The SUP-B15 (ATCC^®^ CRL-1929™) cell line was purchased from American Type Culture Collection (ATCC, Manassas, VA, USA) and was grown in 20% fetal bovine serum (FBS) and 80% Iscove’s modified Dulbecco’s medium with 4 mM L-glutamine adjusted to contain 1.5 g/L sodium bicarbonate and supplemented with 0.05 mM mercaptoethanol. The cell lines were incubated at 37 °C under a 5% CO_2_ atmosphere. For subculturing, cells were grown as non-adherent cell suspensions in T25 culture flasks (Corning, Corning, NY, USA) by maintaining the cell density between 5 × 10^5^ and 2 × 10^6^ cells/mL, with fresh media changes every 2–3 d either via dilution or replacement of the new medium.

### 4.5. Immunophenotyping of Cells

Using a panel of CD10 (clone 212504), CD34 (clone QBEND/10), and TdT (clone TDT-6), we are able to detect immature blast (CLCs) populations in the blood of B-ALL patients after release from the chip (Figure 3C). Staining using an autostainer followed a standard protocol: (i) fixation with 3% buffered PFA (15 min), (ii) staining for surface antigens (CD10 and CD34) (40 min), (iii) permeabilization with 0.1% Triton X-100 (5 min), and (iv) staining for TdT or DAPI (40 min). Washes with 0.05% Tween/PBS were performed after each step.

### 4.6. Sample Preparation for FISH

CLCs were released from the chip into a 5 mL conical tube. Cells were spun down at 400× *g* for 5 min, and the PBS supernatant was removed. Pre-warmed KCl with Colcemid (500 µL) was added to the tube, and the tube was incubated at 37 °C for 5 min. The tube was then spun at 400× *g* for 6 min, and the Colcemid supernatant was removed. Next, 500 µL cold Carnoy’s fixative (3:1 methanol:acetic acid) was added to the tube, and the tube was left to sit on ice for 10 min. The tube was then spun again at 400× *g* for 10 min, and the supernatant was removed. The Carnoy’s fixative wash steps were performed two more times with a final 500 µL of Carnoy’s fixative left in the tube and stored in a −20 °C freezer. The cells were sent to the Children’s Mercy Hospital Genome Core Service Center (GCSC) Genomics Laboratory for FISH analysis.

### 4.7. RNA Isolation

After cells were isolated and the chip was rinsed with 2 mL 0.5% BSA in PBS, the chip was rinsed with 100 µL RNAlater solution at 25 µL/min. Total RNA was extracted using the Direct-zol Microprep kit (Zymo, Irvine, CA, USA) and stored at −80 °C or immediately reverse-transcribed into cDNA. See Appendix A for the total RNA masses isolated.

### 4.8. Reverse Transcription (RT)

Two microliters of purified total RNA was analyzed with a Nanodrop and total RNA quantified. Total RNA was reverse-transcribed into cDNA using the Maxima First Strand cDNA Synthesis Kit for RT-qPCR (Thermo Fisher) with DNAse treatment of RNA and reverse-transcribed using poly(dT) primers according to the manufacturer’s protocol. Negative RTs were performed without an enzyme.

### 4.9. Droplet Digital PCR (ddPCR)

The primers used in gene expression profiling were purchased from Integrated DNA Technologies, IDT DNA (sequences provided in Appendix A). cDNA was used with EVAGreen^®^ Supermix (Bio-Rad, Hercules, CA, USA) for ddPCR. Droplets were formed following the manufacturer’s instructions using the QX200™ Droplet Generator (Bio-Rad). PCR was performed using a C1000 Touch™ Thermal Cycler with a 96-well Fast Reaction Module (Bio-Rad) under the following conditions: 95 °C for 5 min; 40 cycles of denaturation at 95 °C for 30 s, annealing at 53 °C for 30 s, and elongation at 72 °C for 1 min; and a final cooling step at 4 °C. A QX200 droplet reader (Bio-Rad) detected the droplets based on the presence of the product emitting a fluorescent signal. Fluorescent signals were analyzed using QuantaSoft AP_1.0.596 software (Bio-Rad) to generate absolute copy numbers. Copy numbers were normalized to the total RNA (1 ng).

### 4.10. Image Collection and Analysis

A Zeiss Axiovert 200M microscope was equipped with a 10× objective (NA = 0.3), an XBO 75 lamp, DAPI/FITC/Cy3/Cy5 filter sets (Omega Optical, Brattleboro, VT, USA), a Cascade 1K EMCCD (Photometrics) camera, and a MAC 5000 stage (Ludl Electronic Products, Hawthorne, NY, USA), all of which were computer-controlled via Micro-Manager. The automated stage allowed images of the area of interest to be taken to be stitched together. Custom scripts in ImageJ Java 1.8.0_322 (64 bit) (FIJI) were used to both stitch the microscope images together into a final cohesive image with layers for each corresponding dye color and count the number of positive cells for each of the cellular markers. For cell counting, each layer of the stitched images was background subtracted (rolling ball ratio = 50.0 pixels), and the edges of the images were cut and non-cellular debris removed. Next, the intensity threshold on the images was set to maximize the contrast and smoothed to ensure the cells would be detected as circular. The “watershed” function under the “binary” submenu was used to separate cells that were stuck together. After this step, the “analyze particles” function was used with the following parameters: size of 27–314 pixels and circularity of 0.3–1.0. The resulting cell counts were converted into masks for colocalization determination.

Colocalization calculations were performed using ImageJ’s matrix mathematics functions to determine if fluorescence could be detected in multiple channels. Channels were added together and made into new masks that were analyzed for particles as described previously. This resulted in counts for cells that expressed different combinations of markers.

## 5. Conclusions

We presented a microfluidic assay for analyzing peripheral blood of B-ALL patients for the detection of MRD, negating the need for an invasive BMA sample. We addressed the challenge of the rarity of leukemic cells in peripheral blood by using a platform that can enrich the sample for CD19-expressing leukemic cells by affinity isolating the cells of interest using an appropriately designed microfluidic. The microfluidic chip allows enrichment of CLCs using immobilized capture antibodies on the channel’s surface. Once the CLCs are enriched, they could be released from the capture surface and enumerated via immunophenotyping. In addition, the enriched cells were available for molecular profiling, FISH, and genetic testing (i.e., NGS). The MF-MRD based on the affinity isolation of CLCs offers the opportunity for frequent testing to longitudinally track MRD, measure the treatment response, and detect the potential onset of disease recurrence. Microfluidics offer the advantage of the efficient processing of small input blood volumes compared to MFC. We successfully detected MRD from 2 mL of blood, which is particularly important, as low-volume blood draws are a necessity for pediatric patients, especially infants.

MF-MRD detection can be useful for the assessment of the treatment response for classification of a patient into a risk group but also to monitor patient well-being after completion of treatment as a new standard of care for early recognition of impending disease recurrence. In the case of any acute type of disease, such as B-ALL, acute myeloid leukemia, or multiple myeloma, the speed of treatment intervention to disease recurrence is of high value in a medical setting to improve patient outcome.

## Figures and Tables

**Figure 1 ijms-25-10619-f001:**
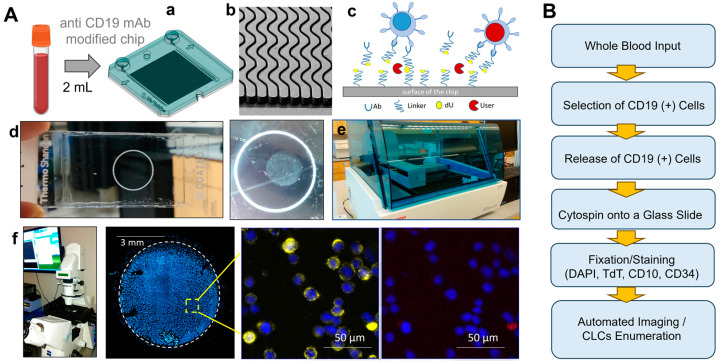
(**A**) Whole blood was processed through microfluidic device modified with mAbs specific for CD19. (**a**) mAb-coated surfaces were false-colored dark green to represent an anti-CD19 mAb-coated device. (**b**) SEM of the sinusoidal channel array (150 channels in the array) and the single channel that addresses all sinusoidal channels. (**c**) Schematic of the affinity isolation assay. CD19(+) antigen-expressing cells bind to surface-tethered mAbs and are retained in the device, while other blood components are passed through the device. Selected cells (fixed or viable) are enzymatically (USER™) released from the capture surface and collected into the cytospin funnel and (**d**) deposited on the glass slide with the aid of cytospin (**e**,**f**). Cells are PFA-fixed and attached to the surface of a poly lysine coated glass slide. Slides are mounted in the autostainer and are immunostained against CD34 and CD10 using fluorescent mAbs, followed by permeabilization and staining for the aberrant marker TdT or DAPI in the nucleus. CLCs are identified by positive aberrant staining (TdT) and CD34/CD10 and DAPI staining, whereas normal cells only show DAPI but not TdT. (**B**) Flow chart showing the entire workflow for processing patient blood samples to search for CLCs.

**Figure 2 ijms-25-10619-f002:**
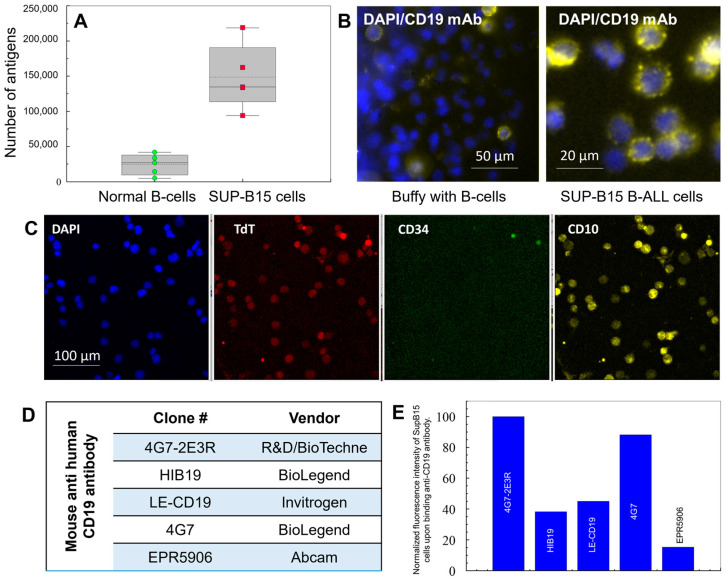
(**A**) Flow cytometry (FC) results for staining of buffy coat cells and the SUP-B15 cell line with anti-CD19 mAb-APC (clone 4G7-2E3R) for the quantification of CD19 receptors. (**B**) Fluorescence images of buffy coat and SUP-B15 cell line stained with anti-CD19 mAb-APC, same clone as in (**A**). (**C**) Immunophenotyping of SUP-B15 cells with nuclear DAPI marker, terminal deoxynucleotidyl transferase (TdT) aberrantly expressed in the nucleus of ALL cells, and CD34 and CD10 surface markers. (**D**) Various anti-CD19 mAbs tested via FC in the SUP-B15 cell line. (**E**) Expression of CD19 in the SUB-B15 cell line for different clones, as determined by FC.

**Figure 3 ijms-25-10619-f003:**
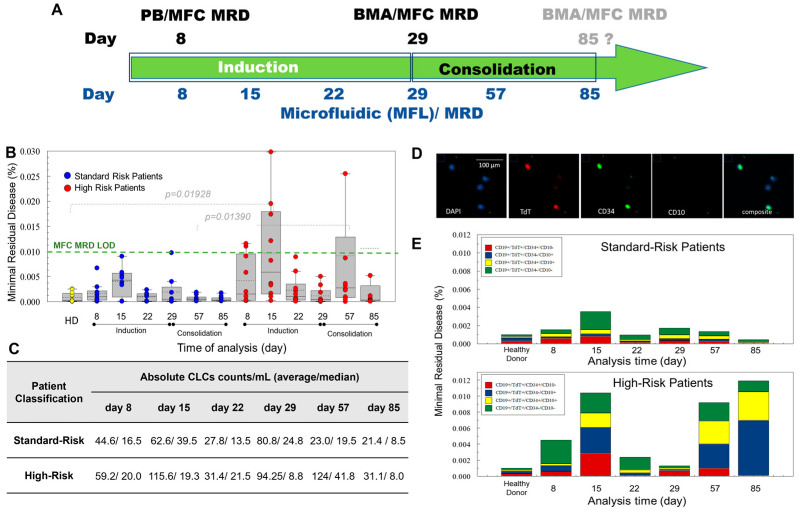
(**A**) Enumeration of B-ALL CLCs in pediatric patients. (**B**) CLCs were isolated from 2 mL blood and their % MRD calculated based on mononucleated WBCs, as determined during hospital administered blood tests. (**C**) Absolute numbers of CLCs detected in mL of blood during longitudinal MRD tracking during treatment (i.e., induction and consolidation). Average and medians are reported in the table. (**D**) Example of immunophenotyped cells and the detection of 2 CLCs, stained with DAPI reagents to identify the nucleus, and the anti-TdT, anti-CD34, and anti-CD10 antibodies, magnification 20x. (**E**) Phenotypes distribution for patients with SR and HR B-ALL during the longitudinal MRD tracking.

**Figure 4 ijms-25-10619-f004:**
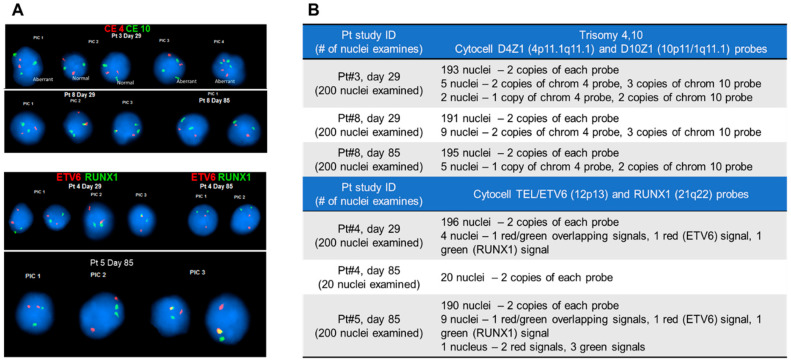
(**A**) Fluorescence images of CD19(+) cells’ nuclei stained with DAPI following FISH testing. (**B**) A summary of the findings for cells tested for Trisomy 4 and 10 and TEL/ETV6 and RUNX1.

**Figure 5 ijms-25-10619-f005:**
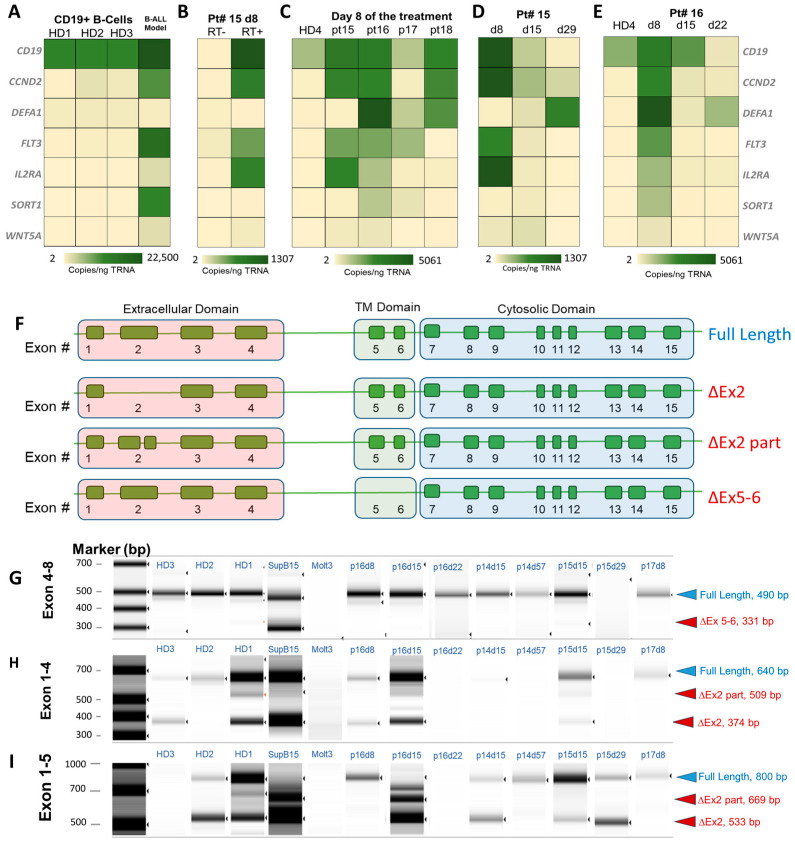
(**A**–**E**) Gene expression profiling of affinity isolated CD19(+) cells using a microfluidics used for MF-MRD detection. Droplet digital PCR (ddPCR) was used to amplify and quantify cDNA reverse-transcribed from isolated total RNA. (**F**) Predicted isoforms of *CD19* mRNA, with deletion of exons 5 and 6 (ΔEx5–6), skipping exon 2 (ΔEx2) and the partial deletion of exon 2 (ΔEx2part) that shifts the reading frame. (**G**–**I**) Gel electropherograms for the separation of amplification products from testing (**G**) exons 4–8, (**H**) exons 1–4, and (**I**) exons 1–5.

**Table 1 ijms-25-10619-t001:** Clinical and analytical figures of merit for MRD methods [11]. *—compared to NGS [12], and **—compared to MFC [19].

Merits	Method for Minimal Residual Disease Detection in B-ALL		
	Multiparameter Flow Cytometry	Quantitative PCR or RT-qPCR	Next Generation Sequencing
**Material tested**	fresh cells	gDNA/mRNA	gDNA
**Target**	ALL immunophenotypes	IG/TCR rearrangement (in Philadelphia—negative B-cell ALL BCR-ABL1 mRNA (Philadelphia positive B-ALL)	IG/TCR gene rearrangements IGH and IGK/IGL rearrangements *****
**Analytical Sensitivity**	10^−4^ (0.01%)	10^−4^–10^−5^ (0.01–0.001%)	10^−6^ (0.0001%)
**Clinical Sensitivity**	61–74% *	≤50% (see challenges)	26–39% more than MFC **
**Advantages**	Least expensive, Can quantify aberrant surface	Sensitive Simple when primers established Standardization available	Very sensitive Can detect different ALL subclones and clonal evolution Standardization available
**Challenges**	Sample preparation needed Lack of standardization between labs, Significant technical expertise needed for 8 color MFC	TCR gene rearrangement not developed in immature blasts designing allele specific oligonucleotides-PCR for each patient is a time-consuming, expensive, and complicated process	gDNA required before treatment start to identify B-ALL clones Expensive Clinical significance of very low levels of MRD not established in all studies *

**Table 2 ijms-25-10619-t002:** B-ALL Pediatric patients’ characteristics at the time of diagnosis.

Patient and Disease Characteristics	n (%)
**Gender**	
Female	9 (45%)
Male	11 (55%)
**Age**	
<1 year-old	0 (0%)
1–10-year-old	17 (85%)
>10-year-old	3 (15%)
**Cytogenetics**	
t(9;22)	0 (0%)
Double trisomies (+4,+10)	7 (35%)
KMT2A rearrangement	0 (0%)
t(12;21)	5 (25%)
iAMP21	0 (0%)
**WBC count**	
<50 × 10^6^/mL	18 (90%)
>50 × 10^6^/mL	2 (10%)
**CNS Involvement**	
Yes	0 (0%)
No	20 (100%)
**Risk Classification after Consolidation Therapy**	
Standard Risk	11 (55%)
High Risk	9 (45%)

**Table 3 ijms-25-10619-t003:** Characteristics of the MF-MRD testing presented herein.

Microfluidic (MF)-MRD Testing	
Characteristic	Metric
Biological Material Tested	Peripheral blood/CD19(+) cells
Blood sample consumption	2 mL for enumeration, 2–5 mL for molecular profiling
Processing throughput	1.5 mL/h
LOD	5 × 10^−4^%
Specificity	85.7%
Sensitivity	71.0%
NPV	100%
Cost per analysis	$130
Molecular analysis possible?	Yes
FISH possible?	Yes
Can observe blasts clonal evolution?	Yes
Clinical significance of low levels of MRD established	No

## Data Availability

Additional experimental details, materials and methods, and supporting results, including figures and graphs, are shown in the Appendix A. All data are available based on request from the readership of this work.

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
