# Peer review of "Microfluidic Affinity Selection of B-Lineage Cells from Peripheral Blood for Minimal Residual Disease Monitoring in Pediatric B-Type Acute Lymphoblastic Leukemia Patients"

_ijms, 2024, doi:10.3390/ijms251910619_

Round 1

Reviewer 1 Report

Comments and Suggestions for Authors

The isolation of rare cells from blood, particularly since the advent of microfluidics for detecting circulating tumor cells, has garnered significant interest. Microfluidic affinity selection for capturing target cells, such as circulating tumor cells, has been extensively studied and has demonstrated considerable potential for clinical use. This study employs microfluidics to isolate and detect B-lineage cells from blood, thereby expanding the scope of microfluidic devices in the isolation of rare cells. Additionally, the authors have validated the feasibility of conducting downstream analyses on the isolated B-lineage cells and have conducted a comparative study between the microfluidic-based test from blood and the standard-of-care assay from bone marrow aspirate (BMA), underscoring the potential clinical utility of the proposed technique. The methodology is logically sound, but further experimental details would enhance clarity.

1. The authors have discussed a microfluidic device designed for isolating circulating tumor cells, which are significantly larger than the B cells targeted in this study. Therefore, the influence of the device's design and dimensions, such as the width, length, and depth of the microchannels, on the isolation efficiency should be analyzed.

2. The authors also noted that shear force could affect the extraction and isolation of rare target cells. This parameter should be analyzed to elucidate its impact on the affinity-based isolation of B cells from blood samples.

3. The functionality of the microchannel coated with CD19 antibody is crucial for achieving high isolation and recovery rates of CD19-positive B cells. Therefore, the immobilization conditions of the CD19 monoclonal antibody (mAb) should be detailed. For example, the EDC/NHS reaction conditions, antibody concentration, and immobilization time used in the study should be specified.

Author Response

Manuscript ID: ijms-3221544

Title: Microfluidic Affinity Selection of B-lineage Cells from Peripheral Blood for Minimal Residual Disease Monitoring in Pediatric B-type Acute Lymphoblastic Leukemia Patients.

Authors: Witek et al.

We sincerely thank the reviewers for the important feedback on our manuscript. Per suggestions, we edited the manuscript and addressed comments to improve the quality of the presented work (text highlighted in “yellow” are modifications made to the original submission). Below please see our responses to the questions. The changes made have been highlighted in yellow in both, main manuscript and SI. Please note that the line numbers have changed owing to added text and tables.

Reviewer 1:

Comments and Suggestions for Authors

The isolation of rare cells from blood, particularly since the advent of microfluidics for detecting circulating tumor cells, has garnered significant interest. Microfluidic affinity selection for capturing target cells, such as circulating tumor cells, has been extensively studied and has demonstrated considerable potential for clinical use. This study employs microfluidics to isolate and detect B-lineage cells from blood, thereby expanding the scope of microfluidic devices in the isolation of rare cells. Additionally, the authors have validated the feasibility of conducting downstream analyses on the isolated B-lineage cells and have conducted a comparative study between the microfluidic-based test from blood and the standard-of-care assay from bone marrow aspirate (BMA), underscoring the potential clinical utility of the proposed technique. The methodology is logically sound, but further experimental details would enhance clarity.

  1. The authors have discussed a microfluidic device designed for isolating circulating tumor cells, which are significantly larger than the B cells targeted in this study. Therefore, the influence of the device's design and dimensions, such as the width, length, and depth of the microchannels, on the isolation efficiency should be analyzed.

Answer:

Thank you for this question. We are happy to explain the framework of the effects of physical dynamics and device architecture on the efficiency of cell selection in sinusoidal microchannels.

The following text will address question 1 and 2 from the reviewer.

In our previous works we presented the effects of physical dynamics and device architecture on the efficiency of affinity selection of cells in sinusoidal microchannels. The process of affinity-selection of cells using sinusoidal chip consists of two phases: (i) initiation of contact between a cell and the mAb bound to the surface and (ii) binding of the rolling cell with surface-bound mAbs. With an increasing linear velocity in the curvilinear channel, the resulting centrifugal forces increase the delivery of cells to the outer channel wall. Inherently, the centrifugal forces have a lesser effect on smaller objects, including cells. For example, the centrifugal forces propel an 8 µm B-cell with a velocity four times slower than they would a 16 µm diameter cell and as a result, smaller cells do not reach the antibody decorated outer channel wall as efficiently as large cells, to allow for cell surface antigens interactions with mAb. While the sinusoidal architecture of channels with a 125 µm radius of curvature, 25 µm channel width, and 150 µm depth provided high recovery (~90%) and high throughput for cells with diameters 15-20 µm such as CTCs, the recovery of smaller cells as reported herein is 45%. Theoretically, the cell recovery could be improved by increasing the linear velocity above the currently used 2 mm/s to increase the centrifugal forces. Unfortunately, this action would shorten the residence time of the cell at the surface bound mAb, and therefore affect the overall binding kinetics of antigen and antibody binding, resulting in lower cell recovery. To increase the efficiency of the cell recovery, the channel’s width could theoretically be decreased as well, however, the consequences of such action would be disadvantageous to the purity, specificity, and throughput of the microfluidic assay. The purity of isolated rare cells would be compromised owing to the presence of larger cells in blood (i.e., granulocytes), that would be physically trapped in channels. Owing to the narrow sinusoidal channels and fast linear velocity of processing, the fluid shear stress for blood generated in the sinusoidal channels is 13.3 dynes/cm2. Such high shear forces prevent “permanent” non-specific interactions with the channel walls, resulting in highly specific affinity isolation.

  1. The authors also noted that shear force could affect the extraction and isolation of rare target cells. This parameter should be analyzed to elucidate its impact on the affinity-based isolation of B cells from blood samples.

Answer: Please see the answer above.

  1. The functionality of the microchannel coated with CD19 antibody is crucial for achieving high isolation and recovery rates of CD19-positive B cells. Therefore, the immobilization conditions of the CD19 monoclonal antibody (mAb) should be detailed. For example, the EDC/NHS reaction conditions, antibody concentration, and immobilization time used in the study should be specified. Already reported.

Answer: We added detailed description of the surface modification protocol. It is presented in SI.  

Reviewer 2 Report

Comments and Suggestions for Authors

Author Response

Manuscript ID: ijms-3221544

Title: Microfluidic Affinity Selection of B-lineage Cells from Peripheral Blood for Minimal Residual Disease Monitoring in Pediatric B-type Acute Lymphoblastic Leukemia Patients.

Authors: Witek et al.

We sincerely thank the reviewers for the important feedback on our manuscript. Per suggestions, we edited the manuscript and addressed comments to improve the quality of the presented work (text highlighted in “yellow” are modifications made to the original submission). Below please see our responses to the questions. The changes made have been highlighted in yellow in both, main manuscript and SI. Please note that the line numbers have changed owing to added text and tables.

Reviewer 2:

The manuscript entitled “Microfluidic Affinity Selection of B-lineage Cells from Peripheral Blood for Minimal Residual Disease Monitoring in Pediatric B-type Acute Lymphoblastic Leukemia Patients” proposed a microfluidic (MF)-based assay for monitoring of minimal residual disease (MRD) from the blood of pediatric patients via affinity enrichment and immunophenotyping of circulating leukemia cells (CLCs). In contrast to invasive bone marrow aspirates (BMA) sampling, the use of peripheral blood for MRD testing indeed carried important benefits especially for pediatric patients. Moreover, the challenge involving the rarity of leukemic cells in peripheral blood was addressed by using a platform for sample enrichment of CD19-expressing leukemic cells. Although the proposed method would be useful for stratification of patients into risk groups and early recognition of potential impending disease recurrence, some issues still needed to be clarified before acceptance. All comments are listed as below.

Major Point

  1. Actually, the authors paid much effort on the development of the cell selection MF for the isolation of rare circulating tumor cells (CTCs) [27,28] and acute myeloid leukemia (AML) [30]. In other words, similar methods involving a cell selection MF with a series sinusoidal channels, cell staining, and image analysis with the aid of epifluorescence microscope and the developed software have been proposed elsewhere by the same research group. Perhaps the authors should indicate the novelty of this study in terms of “Technique Development” compared to that of the study reported in Analyst [30]. For example, the benefits of using the Cytospin compared to a 96-well plate.

Answer: We expanded the following paragraph to accommodate requested edits.

Following the release of the selected cells from the surface, eluent was collected in an assembled cytospin funnel with a lysine modified glass slide and was cytospun using a centrifuge, which automated sample processing when compared with our previously reported assay. We improved the processing pipeline and developed more optimized methodology compared to our previous work, where the captured AML CLCs were immunostained on-chip, manually released into a 96 well plate and visualized/enumerated using fluorescence microscopy. This workflow required manual handling of cells and placement into the wells of the titer plate. Unfortunately, the assay lacked automation making it prone to potential loss of cells and/or reproducibility issues due to operator dependent CLC handling.

  1. Although significant improvement of the limit-of-quantitation (LOQ) and MRD(+) determination was observed in this study, the related information of the analytical results obtained by the previous study (e.g., throughput, cost per analysis, sample consumption, and so on) should be added for objective assessment of the superiority of the method developed in this study. A comparative table outlining the characteristics of the proposed method against existing methods should be included.

Answer: Thank you for that suggestion. Summarized characteristics of MF-MRD will be presented in Table 3.

Table 3. Characteristics of the MF-MRD Testing presented herein.

Microfluidic (MF)-MRD Testing

Characteristic

Metric

Biological Material Tested

Peripheral blood/ CD19(+) cells

Blood sample consumption

2 mL for enumeration

2-5 mL for molecular profiling

Processing throughput

1.5 mL/h

LOD

5 × 10-4 %

Specificity

85.7%

Sensitivity

71.0%

NPV

100%

Cost per analysis

$130

Molecular analysis possible?

Yes

FISH possible?

Yes

Can observe blasts clonal evolution?

Yes

Clinical significance of low levels of MRD established

No

  1. As described by the authors, sampling of peripheral blood may be also more accurate as blood is homogenous medium unlike BMA due to its non-uniform spatial composition, which can lead to sampling errors. However, the repeated analyses of samples collected from same patients achieved by the developed method was absent. In other words, the precision of the developed method was not validated yet.

Answer: We included the following:

For selected patients (n=3), we performed MF-MRD analysis in duplicate, and we observed a chip-to-chip reproducibility of ~12% ensuring accuracy of the MRD test.

  1. Detailed information of the statistical method used to determine the LOD should

be provided instead of brief descriptions (line 608).

In the results section (page 281) we described how the LOD were established.

To establish the LOD for the MF-MRD, we tested healthy donors’ blood using the same protocol for testing B-ALL patient’s blood (Table S4). The average nucleated cells that were TdT(+) was 8.8 ±5.4/mL. The threshold was established as the average number plus 3× SD of detected TdT(+) cells in healthy donor blood, which was 25/mL of blood containing ~5 × 106 mononuclear WBCs. Therefore, the LOD was established as 25 cells/5 × 106 cells (i.e., 5 × 10-4 %). We concluded that above the 5 × 10-4 % threshold, MRD was considered positive (Table S5 and S6).

Minor Points

Answer: We apologize for those mistakes that occurred when manuscript was introduced to the template. Thank you very much for noticing those mistakes. All and others were corrected.

  1. Several representations should be refined. For example, “a threshold of 5 x 10-4

% (line 26)”, “Constructed circular (d=28 mm2) (line 175)”, “platelet count ≥100

× 109/L (line 260)”, “platelet count <100 × 109/L (line 263)”, “interrogate ~5 ×

106 cells (line 521)”, “interrogating 4 × 106 BMA cells (line 523)”, “patients with

MRD <10−4 (line 600)”, “leukemic blasts ranging 10−4 to 10−3 (line 601)”, and

so on.

  1. Some mistakes in the main text should be corrected. For example, “…was

negative for CD34 (Fig. 1B, C) (line 204)”; “…for the selection of CLCs (Fig. 1D,

  1. E) (line 205)”; “…incubation with clone 4G7-2E3R as shown by our flow

cytometry data, (Fig. 1E) (line 207)”.

  1. Some figure captions and table footnote should be refined. For example, the term

“na” shown in Figure S1 (B).

Answer: We defined acronyms in tables.
